# Clinical, Genetic and Functional Characterization of a Novel *AVPR2* Missense Mutation in a Woman with X-Linked Recessive Nephrogenic Diabetes Insipidus

**DOI:** 10.3390/jpm12010118

**Published:** 2022-01-17

**Authors:** Senthil Selvaraj, Dírcea Rodrigues, Navaneethakrishnan Krishnamoorthy, Khalid A. Fakhro, Luís R. Saraiva, Manuel C. Lemos

**Affiliations:** 1Department of Genetics, Sidra Medicine, Doha P.O. Box 26999, Qatar; sselvaraj@sidra.org (S.S.); nkrishnamoorthy2@sidra.org (N.K.); kfakhro@sidra.org (K.A.F.); 2Serviço de Endocrinologia, Diabetes e Metabolismo, Centro Hospitalar Universitário de Coimbra, 3000-075 Coimbra, Portugal; dircearodrigues@chuc.min-saude.pt; 3Department of Genetic Medicine, Weill Cornell Medical College, Doha P.O. Box 24144, Qatar; 4College of Health and Life Sciences, Hamad Bin Khalifa University, Doha P.O. Box 24144, Qatar; 5Monell Chemical Senses Center, 3500 Market Street, Philadelphia, PA 19104, USA; 6CICS-UBI, Health Sciences Research Centre, University of Beira Interior, 6200-506 Covilhã, Portugal

**Keywords:** vasopressin, AVPR2, nephrogenic diabetes insipidus, genetics, case report

## Abstract

Nephrogenic diabetes insipidus (NDI) is a rare disorder characterized by renal unresponsiveness to the hormone vasopressin, leading to excretion of large volumes of diluted urine. Mutations in the arginine vasopressin receptor-2 (*AVPR2*) gene cause congenital NDI and have an X-linked recessive inheritance. The disorder affects almost exclusively male family members, but female carriers occasionally present partial phenotypes due to skewed inactivation of the X-chromosome. Here, we report a rare case of a woman affected with X-linked recessive NDI, presenting an average urinary output of 12 L/day. Clinical and biochemical studies showed incomplete responses to water deprivation and vasopressin stimulation tests. Genetic analyses revealed a novel heterozygous missense mutation (c.493G > C, p.Ala165Pro) in the *AVPR2* gene. Using a combination of in-silico protein modeling with human cellular models and molecular phenotyping, we provide functional evidence for phenotypic effects. The mutation destabilizes the helical structure of the AVPR2 transmembrane domains and disrupts its plasma membrane localization and downstream intracellular signaling pathways upon activation with its agonist vasopressin. These defects lead to deficient aquaporin 2 (AQP2) membrane translocation, explaining the inability to concentrate urine in this patient.

## 1. Introduction

Water homeostasis in the body is maintained by balancing thirst-stimulated water intake and renal water excretion. Urine volume and concentration is regulated by the action of vasopressin (or antidiuretic hormone), which is secreted into the bloodstream by the posterior lobe of the pituitary gland in response to extracellular fluid hypertonicity [1]. Deficiencies in vasopressin secretion or renal unresponsiveness to vasopressin lead to central diabetes insipidus or nephrogenic diabetes insipidus (NDI), respectively [2]. Both diseases are characterized by excessive urine production (polyuria) and water intake (polydipsia). An inappropriately dilute urine with a high serum osmolality, assessed through a water-deprivation test, establishes the diagnosis of diabetes insipidus [2]. NDI can be further distinguished from central diabetes insipidus by the failure to normalize urine osmolality upon stimulation with the vasopressin synthetic analogue, desmopressin (1-deamino-8-D-arginine vasopressin, dDAVP) [3].

Congenital NDI is a rare disorder that is usually inherited as an X-linked recessive trait due to mutations in the arginine vasopressin receptor 2 (*AVPR2*) gene [4,5,6] or, less frequently, as an autosomal recessive or dominant trait due to mutations in the aquaporin-2 (*AQP2*) gene [7]. These genetic forms of NDI are usually present since birth and can result in life-threatening dehydration and neurologic impairment if water intake is inadequate [8,9]. The *AVPR2* gene is located on the X-chromosome (region Xq28), consists of three coding exons and encodes a 371-amino acid G protein-coupled receptor (GPCR) that contains seven transmembrane domains [10,11]. *AVPR2* is expressed at the basolateral membrane of the principal cells of the collecting duct in the kidneys, where water reabsorption takes place. The binding of vasopressin to AVPR2 in the kidney activates adenylyl cyclase, increasing intracellular cyclic adenosine monophosphate (cAMP), which in turn activates cAMP-dependent protein kinase. This activation leads to the phosphorylation and trafficking of the AQP2 channel, followed by insertion of AQP2 along the apical cell membrane of the collecting duct, thereby allowing water to enter the cell and reducing renal excretion of water [7].

To date, at least 280 different loss-of-function mutations in *AVPR2* have been reported in patients with NDI (Human Gene Mutation Database, www.hgmd.org, accessed on 5 January 2022). In vitro functional studies have shown that mutant receptors are either improperly transcribed, retained in the endoplasmic reticulum, or reach the cell surface, but are unable to properly bind vasopressin or to trigger an intracellular cAMP signal [12]. As the mode of inheritance is X-linked recessive, affected individuals are mainly males who are hemizygous for the mutations. While heterozygous females are usually unaffected, some females may exhibit variable degrees of polyuria and polydipsia due to skewed inactivation of the X-chromosome [13].

This study aimed to investigate the genetic defect and its functional consequences in a woman affected with NDI. Using targeted sequencing analysis, we identified a novel heterozygous missense mutation in *AVPR2,* which causes a substitution of the alanine (Ala) to proline (Pro) at residue position 165 (Ala165Pro). We then tested the impact of this mutation on the protein structure using an in-silico approach and assessed its functional consequences by using in vitro human cell models.

## 2. Material and Methods

### 2.1. Clinical Studies

A 26-year-old Portuguese woman presented a history of polyuria and polydipsia. She reported that her symptoms had been present since infancy. Despite significant polyuria (approximately 12 L/day), she had adapted to her long-standing condition and never sought medical treatment. Her past medical history had no other relevant occurrences. Family history included four affected males (father, great-uncle, uncle and cousin) (Figure 1A), who declined investigation. Physical examination was unremarkable. We performed a water-deprivation and dDAVP stimulation test for the differential diagnosis of diabetes insipidus [2,3]. Starting in the morning, all fluid intake was stopped and body weight, urine output and plasma and urine osmolarity were measured hourly until the patient lost 5% of her body weight. At this time, fluid intake was resumed to prevent further dehydration and a 2 μg intramuscular injection of dDAVP was administered, followed by hourly measurements of urine osmolarity. The cardiovascular and coagulation responses to dDAVP were determined on a different day [14]. An intravenous line with a three-way stopcock was inserted in one arm while the patient was supine and baseline blood samples collected at −30 and 0 min. The patient then received an infusion of dDAVP (0.3 μg/kg, in 100 mL saline, over 20 min) and additional blood samples collected at 30, 60, 90 and 120 min. Blood pressure and heart rate were recorded every 10 min from −30 to 0 min, every 5 min from 0 to 30 min and every 10 min from 30 to 120 min. Blood samples were centrifuged immediately after collection and the plasma separated for measurements of factor VIIIc, von Willebrand factor and renin activity [14].

### 2.2. Genetic Studies

The genetic studies were approved by the Institutional Research Ethics Committee of the Faculty of Health Sciences at the University of Beira Interior, Portugal (study reference number CE-FCS-2013-017) and written informed consent obtained from the patient. Genomic DNA was extracted from peripheral blood leukocytes of the patient and used with polymerase chain reaction (PCR) primers to amplify the coding regions of the *AVPR2* gene (primer sequences and PCR conditions are available upon request). We performed bidirectional sequencing of the PCR products using an automated capillary DNA sequencer (GenomeLab TM GeXP, Genetic Analysis System, Beckman Coulter, Fullerton, CA, USA) and the identified genetic variant was validated by restriction enzyme analysis with *Mwo*I. The parents of the patient accepted genetic testing and were screened for the presence of the variant by restriction enzyme analysis. To determine if the identified genetic variant was present in the general population [15], we searched the Genome Aggregation Database (gnomAD) [16] and the 1000 Genomes Project. Bioinformatic programs (PolyPhen-2, SIFT, PROVEAN and MutationTaster) [17] were used to predict the consequences of the variant on protein function. The variant was analyzed by VarSome [18] and classified according to American College of Medical Genetics and Genomics (ACMG) criteria [19]. We based the nomenclature of the variant on the *AVPR2* cDNA reference sequence (GenBank accession number NM_000054.6), whereby nucleotide c.1 was the A of the ATG-translation initiation codon.

### 2.3. Molecular Modeling of AVPR2

As the human AVPR2 lacks a 3D structure from experimental studies (such as X-ray crystallography or nuclear magnetic resonance), we performed the molecular modeling of its protein 3D structure to assess the impact of the Ala165Pro substitution. Previously reported modeling of AVPR2 was based on the amino acid replacement method with another computer model of AVPR1 [20,21]. After the release of this model, the X-ray structures and structure-function studies of multiple other related G protein-coupled receptors (PDB ID: 5WQC, 6TQ4, 6TP3, 6U1N, 6TP6, 6TPK, 6TPJ, 6V9S) were released [22,23,24,25,26,27]. We chose the X-ray structure of the oxytocin receptor (PDB ID: 6TPK) as the template for the AVPR2 model building due to its sequence and structural conservation (44% identity, 58% similarity), similar fold, domain distribution and shared key binding sites.

We used the template specification method to model the human AVPR2 in Swiss-model [28]. The quality of the model was estimated using the recently developed distance constraints method [29]. Structural analyses, secondary structural observations, binding regions and GPCR-family such as conserved residues were mapped using PyMOL (The PyMOL Molecular Graphics System, Schrödinger LLC, New York, NY, USA) as previously described [30]. The structure of the Ala165Pro-AVPR2 mutant was predicted using the structure generated with the wild-type AVPR2 sequence as a template and local network analysis of the intra-molecular interactions in the wild-type and Ala165Pro substitution was carried out as previously described [31].

### 2.4. Cell Culture and Transfection

HEK293 cells (ATCC) were grown in Gibco Dulbecco’s Modified Eagle Medium (DMEM, Thermo Fischer Scientific, Waltham, MA, USA) containing 10% fetal bovine serum (FBS, Thermo Fischer Scientific, Waltham, MA, USA), 100 U/mL penicillin and 100 μg/mL streptomycin at 37 °C in a humidified atmosphere with 5% CO2. Plasmids pcDNA 3.1 (+) WT-AVPR2-GFP, A165P-AVPR2-GFP, WT-AVPR2-HA, A165P-AVPR2-HA and AQP2 were purchased from Genscript Inc (Piscataway, NJ, USA). For the transient transfection, we used Lipofectamine 3000 (Thermo Fischer Scientific, Waltham, MA, USA) according to the manufacturer’s instructions. At 36 h after transfection, the cells were used for the downstream analysis.

### 2.5. cAMP and PKA Activity Assays

The level of cAMP in AVPR2 transfected HEK293 cells was measured using a cAMP direct immunoassay kit as described by the manufacturer’s instructions (Abcam, Cambridge, UK). Briefly, the day before transfection, HEK293 cells were plated in a 6-well plate at around 70–80% confluency. Cells were transfected with WT-AVPR2 or A165P-AVPR2 at the concentration of 2.5 μg per well using Lipofectamine 3000. At 36 h after transfection, cells were stimulated with or without AVPR2 agonist dDAVP. Cells were washed with PBS, lysed using 0.1 M HCl and the cell lysates were centrifuged at 14,000× *g* for 10 min. The supernatant was used to measure the cAMP levels and the ELISA plate was read at OD 540 nm.

We measured the cellular activity of PKA using a non-radioactive PKA kinase assay kit according to the manufacturer’s instructions (Abcam, Cambridge, UK). PKA measurement in cells was based on a solid phase enzyme-linked immuno-absorbent assay (ELISA) that utilizes a specific synthetic peptide as a substrate for PKA and a polyclonal antibody that recognizes the phosphorylated form of the substrate. HEK293 cells were transfected with WT-AVPR2 or A165P-AVPR2 in a 6-well plate as described above. Cells were lysed with lysis buffer and the activity of PKA was measured at 450 nm.

### 2.6. Western Blotting and Cell Surface Biotinylation Assay

We used the Mem-PER Plus Membrane Protein Extraction Kit (Thermo Fischer Scientific, Waltham, MA, USA) to isolate the crude membrane from cells expressing WT and A165P-AVPR2. Briefly, cells were treated with mild detergent to solubilize the cytosolic protein. The cytosolic fraction was isolated by centrifugation and the remaining cell pellet was treated with a second detergent to solubilize the membrane protein and centrifuged to collect the membrane fraction. Cell surface biotinylation was performed using the Pierce Cell Surface Biotinylation and Isolation Kit (Thermo Fischer Scientific, Waltham, MA, USA) according to the manufacturer’s instruction. Briefly, the surface proteins were labeled by incubating the cells with EZ-Link Sulfo-NHS-SS-Biotin at 4 °C. The cells were lysed and the labeled protein with debris removed by centrifugation (12,000× *g*, 15 min, 4 °C). Supernatant fractions (500 µL) were incubated with protein-A Sepharose, preincubated with rabbit polyclonal anti-HA antibody and incubated overnight at 4 °C. Beads were washed twice with 250 µL of lysis buffer. The crude membrane and cell surface biotinylated samples were resolved in 4–12% Nupage gel and transferred into a Polyvinylidene Fluoride (PVDF) membrane. The PVDF membrane was blocked with non-fat milk buffer and incubated overnight with respective primary antibody at 4 °C, followed by HRP-conjugated secondary antibody incubation. The resolved proteins were visualized using Chemdoc (Bio-Rad Laboratories, Hercules, CA, USA) and bands were quantified by densitometric analysis using ImageJ (NIH, Bethesda, MD, USA), as previously described [32].

### 2.7. Immunofluorescence

Immunofluorescence was used to visualize the cell surface expression of GFP-tagged AVPR2. HEK293 cells were transfected either with GFP tagged- WT-AVPR2 or A165P-AVPR2. Cell surface receptor staining was performed on live cells by incubation with a rabbit polyclonal anti-GFP (Thermo Fischer Scientific, Waltham, MA, USA) primary antibody diluted in DMEM with 1% BSA) for 1 h at 4 °C. Cells were then washed with ice-cold phosphate-buffered saline (PBS) and fixed with 3% PFA for 15 min at room temperature followed by incubation with a Alexa Fluor 546-conjugated (Thermo Fischer Scientific, Waltham, MA, USA) secondary antibody.

### 2.8. Statistical Analysis

Data analysis was performed using GraphPad Prism 8.0.0 software (GraphPad, San Diego, CA, USA). Statistical comparisons were made using unpaired *t*-tests with Welch’s correction (two-tail) or one-way ANOVAs (Tukey multiple comparisons corrections). Experimental values are displayed as mean ± SEM. Differences in the mean values were considered to be significant at *p* < 0.05.

## 3. Results

### 3.1. Blood and Urine Biochemical Tests

To confirm the diagnosis of NDI, we performed a water-deprivation and dDAVP stimulation test (see Methods). The basal urine output of the patient was 12.3 L/day, with specific gravity 1.003 (normal 1.010–1.030) and osmolality 98 mOsm/kg (normal range 300–900). Basal serum chemistry results, including electrolytes, were normal. The water-deprivation test was stopped after 6 h when the patient lost 5% of her body weight (Figure 1B,C). At this time, the serum osmolality had increased from 276 to 311 mOsm/Kg (normal range: 275–295 mOsm/Kg) and the urine osmolality had increased from 95 to only 191 mOsm/Kg (normal range: 300–900 mOsm/Kg) (Figure 1C). To test the response to vasopressin, we administered a 2 μg intramuscular injection of dDAVP at hour 6 and repeated 4 h later. No increase in urine osmolality was observed (Figure 1C), consistent with renal unresponsiveness to vasopressin. We also determined the cardiovascular and coagulation responses to an intravenous infusion of dDAVP (see Methods) [2,3]. Mean arterial blood pressure decreased 18% (normal > 10%), heart rate increased 10% (normal > 20%), plasma renin activity increased 4% (normal > 65%), factor VIIIc increased 2.2× (normal > 3×), von Willebrand factor increased 1.5× (normal > 2×), thus, showing only partial responses to dDAVP (Figure 1D). To confirm the resistance to vasopressin, the patient was treated with increasing doses of dDAVP, which did not significantly improve the polyuria (Figure 1E). Finally, the patient was put on treatment with hydrochlorothiazide 50 mg bid, amiloride chloridrate 5 mg bid and low-sodium diet, resulting in a reduction of diuresis to approximately 5 L/day (Figure 1E).

### 3.2. Identification of a Novel Missense Mutation in AVPR2

Next, we performed DNA amplification and sequencing of the *AVPR2* gene in the patient (Figure 1F). We identified a heterozygous missense variant in exon 2, which predicted the substitution of an Alanine by a Proline at amino acid position 165 of the protein (chrX:153,171,453 G > C; NM_000054.6: c.493G > C; p.Ala165Pro) (Figure 2B). Restriction enzyme analysis with *Mwo*I confirmed the presence of the variant in the patient and in her father and its absence in her mother (Figure 1G). The variant was absent in both the gnomAD and the 1000 Genomes Project population databases and was predicted to be damaging by multiple in silico prediction tools (PolyPhen-2, SIFT, PROVEAN and MutationTaster). The variant fulfilled the ACMG criteria for “Likely Pathogenic” (PM1, PM2, PP2, PP3).

Taken together, these results confirm the diagnosis of X-linked recessive NDI in this female patient and identified a novel *AVPR2* missense mutation (Ala165Pro) involved in this disease.

### 3.3. The Missense Ala165Pro Variant Disrupts AVPR2 Function

Using molecular modeling tools (see Methods), we first assembled the model structure of the ‘wild-type’ human AVPR2 (Ala165), which includes the seven transmembrane domains/helices (H1–7) connected with three intracellular (I1–3) and extracellular (E1–3) loops, typical of GPCRs (Figure 2A). The Ala165 residue sits in the middle of helix 4, surrounded by the vasopressin binding pocket (Figure 2B), which includes a key GPCR signature residue (Trp164) [22,33]. Helix 4 also contains other residues (Lys100, Ala110, Met120, Leu175, Arg202, Phe307) suggested to play a role in shaping the vasopressin binding pocket and its affinity to its ligand [20] (Figure 2C).

Next, we generated the model structure of the ‘mutant’ human AVPR2 (Figure 2D). The mutant Ala165Pro substitution is in the middle of a block of ten hydrophobic residues (Val Leu Val Ala Trp *Ala* Phe Ser Leu Leu Leu) that provide stability to helix 4 and results in side-chain chemical structure from hydrophobic (Ala) to an uncharged polar ring (Pro) which can introduce a break in the hydrophobicity. To test this assumption, we performed an analysis of the local network of intra-molecular interactions connected with the residue at position 165. We identified a hydrogen bond between Ala165 and Leu161 in the wild-type AVPR2 (Figure 2E, left panel), which does not exist between Pro165 and Leu161 in the mutant AVPR2 (Figure 2F, right panel). The lack of this interaction disconnects the network of 165-161-157 in helix 4, potentially destabilizing the helical structure. Indeed, Ala to Pro mutations in transmembrane helices were previously shown to break and/or distort the helical structure [34,35]. Moreover, another AVPR2 missense substitution in position 165, Ala165Asp, has previously been reported to interfere with proper receptor trafficking [36].

Together, these results reveal that the Ala165Pro missense mutation disrupts the intramolecular interactions and destabilizes the helical structure, which possibly affects AVPR2 trafficking, stability and function.

To further investigate the functional effects of the Ala165Pro substitution on AVPR2, we performed a series of in vitro cellular assays. In these assays, we generated HA-tagged and GFP-tagged versions of AVPR2 containing the wild-type (WT-AVPR2) and the missense mutation (Ala165Pro-AVPR2) identified in the patient and expressed them in HEK293 cells. We found no significant differences between the total protein levels of the wild-type and mutant AVPR2 (Figure 3A). In contrast, the amount of AVPR2 protein present in the crude plasma membrane extracts was significantly reduced in the mutant compared to the wild-type (Figure 3B). Consistent with these results, cell surface biotinylation experiments revealed that the cell surface expression of AVPR2 was also significantly impaired in the mutant compared to the wild-type (Figure 3C). Immunofluorescence experiments provided the third line of evidence for reduced membrane levels of the mutant AVPR2 compared to the wild-type (Figure 3D).

Taken together, these data show that the Ala165Pro mutation does not compromise overall receptor abundance but reduces the plasma membrane levels of AVPR2.

### 3.4. Biological Impact of the Mutant AVPR2

Vasopressin activates the basolateral membrane of AVPR2 in renal collecting duct principal cells, leading to the activation of the Gαs-cAMP-PKA signaling cascade and the subsequent phosphorylation and membrane translocation of AQP2 water channels to initiate water reabsorption [37]. To understand whether the Ala165Pro substitution in AVPR2 affect its function, we first evaluated its ability to generate cAMP and PKA kinase activation in response to dDAVP compared to the wild-type. Upon stimulation with dDAVP, we observed significant increases in the levels of cAMP and PKA kinase activation in cells transfected with WT-AVPR2 and Ala165Pro-AVPR2, but these increases were significantly higher in the wild-type compared to the mutant (Figure 4A,B).

Stimulation of AVPR2 by dDAVP activates the PKA kinase by phosphorylating it at Thr197 and inhibiting the phosphorylation of p38-MAPK [37]. Our experiments further revealed that dDAVP stimulation increased PKA kinase phosphorylation in cells transfected with WT-AVPR2 and Ala165Pro-AVPR2, but these increases were only significant in the wild-type (Figure 4C). In contrast, we observed increased p38-MAPK phosphorylation in cells transfected with the mutant Ala165Pro-AVPR2 compared to the WT-AVPR2.

Next, to study the impact of A165P mutant on AQP2 membrane translocation, we performed cell surface biotinylation experiments after co-transfection with AQP2 and the wild-type or mutant versions of AVPR2. These experiments revealed that upon stimulation with dDAVP, the plasma membrane translocation of AQP2 was significantly increased in WT-AVPR2 but not in the mutant Ala165Pro-AVPR2 (Figure 4E).

Taken together, these data show that the Ala165Pro mutation causes a decrease in cAMP levels and PKA kinase activation in response to dDAVP and subsequently impairs the translocation of AQP2 to the plasma membrane.

## 4. Discussion

Here, we identify and functionally characterize a novel *AVPR2* missense mutation found in a woman with a history of polyuria and polydipsia. We show that the Ala165Pro substitution in helix 4 of AVPR2 impairs the trafficking, stability and function of AVPR2 and the subsequent translocation of AQP2 to the plasma membrane.

Women who are carriers of *AVPR2* mutations are not usually affected by NDI, given the X-linked recessive mode of inheritance. However, in rare cases, this disorder may manifest itself due to a mechanism of skewed X-inactivation [38]. X-chromosome inactivation is a natural process that occurs in every female during early embryonic development, by which each cell randomly inactivates either the paternal or the maternal X-chromosome. As a result, females are usually made up of mosaics of cells that statistically will have half of the active X-chromosomes being paternal and the other half maternal. However, this inactivation will sometimes deviate significantly from this 50:50 distribution. If the predominantly expressed X-chromosome happens to have a recessive mutation, then the woman may exhibit a variable phenotype depending on the percentage of cells expressing the mutant gene in the target organ [38]. X-inactivation patterns can differ between tissues and this can explain why this female patient had NDI yet had partial hemodynamic and coagulation responses to AVPR2 stimulation. These findings suggest that the patient had skewed X-inactivation in the kidneys, whereas a more typical random distribution occurred in the liver and vascular endothelium, where factor VIIIc and von Willebrand factor are produced [39]. Unfortunately, we could not experimentally confirm this in our patient due to the inability to obtain tissue samples from these locations.

In terms of mechanism, our cellular assays revealed that the mutant receptor is translated at levels similar to the wild-type, but it fails to reach the cell surface. These results are consistent with the impaired intracellular transport and cell surface expression of AVPR2 caused by other missense mutations (Trp164Ser, Ala165Asp, Ser167Leu, Ser167Thr, Leu170Pro and Gln174Arg) located in the same region as Ala165Pro [36,40,41,42]. These and other in vitro studies suggest that mutant receptors with distorted transmembrane domains are misfolded, resulting in their retention in the endoplasmic reticulum and proteasomal degradation [12]. On the other hand, defects in the plasma membrane trafficking of AVPR2 mutants result in lower intracellular cAMP levels in recombinant cells [43,44]. However, how changes in the downstream signaling pathways of AVPR2 activation contribute to NDI development is not well established. Thus, understanding the functional consequences of specific mutations may offer new therapeutic avenues for precision medicine approaches. For example, small molecular chaperones have been shown to rescue cell-surface expression and function of some misfolded AVPR2 mutants [45] and cGMP phosphodiesterase inhibitors induce AQP2 membrane insertion independently of the AVPR2 signaling pathways [46].

Agonist mediated stimulation of AVPR2 induces the cAMP signaling cascade in renal collecting duct principal cells that trigger the translocation of AQP2 to the apical membrane, resulting in water reabsorption. In our experiments, the mutant Ala165Pro-AVPR2 showed reduced intracellular cAMP levels and cAMP-mediated PKA activation upon stimulation with dDAVP, consistent with previous studies [47,48,49]. Our experiments further revealed that in the WT-AVPR2, but not in the Ala165Pro-AVPR2, this cAMP-mediated PKA activation leads to a PKA phosphorylation at Thr197 and subsequent inhibition of p38-MAPK and decreased Ser261 phosphorylation, in line with previous results in fibroblasts and HeLa cells [50]. In contrast, in the Ala165Pro-AVPR2, cAMP-mediated PKA activation upon dDAVP stimulation leads to an increase in p38-MAPK phosphorylation, suggesting that production of intracellular cAMP controls the downstream activation of PKA and PKA dependent p38-MAPK inhibition.

Vasopressin regulates AQP2 protein abundance by increasing the intracellular level of cAMP, which activates PKA-mediated inhibition of p38-MAPK and p38-MAPK–dependent regulation of the proteasomal degradation of AQP2 [51,52]. Moreover, vasopressin increases the abundance of AQP2 in the plasma membrane by regulating the trafficking of AQP2 between the intracellular vesicles and the apical plasma membrane. Our experiments show that the Ala165Pro-AVPR2 substitution leads to a reduction in the membrane levels of AQP2 in response to dDAVP compared to the WT-AVPR2. These data suggest that the impaired vasopressin-dependent cAMP signaling leads to defects in AQP2 transcription, translation, stability and trafficking to the membrane, which ultimately manifests in the patient as an inability to concentrate urine.

In summary, we identified a rare case of a woman affected with NDI and showed that her condition was caused by a novel missense mutation of the *AVPR2* gene, leading to impaired trafficking, stability and function of AVPR2. Our study extends the known mutational spectrum of the *AVPR2* gene, the key role of X-inactivation mosaicism in NDI in female patients and further contributes to the understanding of the pathogenesis of this disorder.

## Figures and Tables

**Figure 1 jpm-12-00118-f001:**
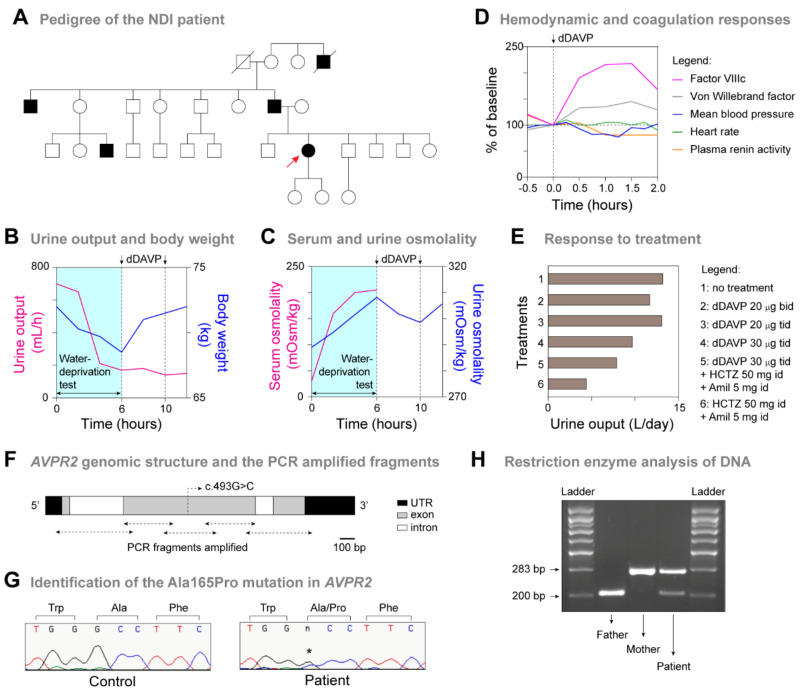
Biochemical and genetic studies of the patient with nephrogenic diabetes insipidus (NDI). (**A**) Pedigree of the patient with NDI. The inheritance follows a typical X-linked recessive pattern, apart from the reported female patient (red arrow). Individuals are represented as males (squares), females (circles), affected (filled symbol), unaffected (open symbol) and deceased (oblique line through symbol). (**B**,**C**) Water-deprivation and dDAVP stimulation tests. The water-deprivation test was continued until the patient lost 5% of her body weight, which occurred after 6 h (**B**). At the end of the test, serum osmolality had increased from 276 to 311 mOsm/Kg (normal 275–295) and the urine osmolality had increased from 95 to only 191 mOsm/Kg (normal 300–900) (**C**). Arrows indicate the administration of dDAVP (2 μg intramuscular injection), which did not increase urine osmolality, thereby supporting the diagnosis of NDI. (**D**) Extra-renal responses to dDAVP. Arrow indicates the administration of dDAVP (0.3 μg/kg, intravenous infusion over 20 min). Mean arterial blood pressure decreased 18% (normal > 10%), heart rate increased 10% (normal > 20%), plasma renin activity increased 4% (normal > 65%), factor VIIIc increased 2.2× (normal > 3×), von Willebrand factor increased 1.5× (normal > 2×). (**E**) Response to treatment. Treatment with increasing doses of dDAVP did not significantly improve the polyuria. Treatment with hydrochlorothiazide (HCTZ) 50 mg bid, amiloride chloridrate (Amil) 5 mg bid and a low-sodium diet, resulted in a reduction of diuresis to approximately 5 L/day. (**F**) Genomic structure (Xq28) showing the exons (grey boxes), introns (white boxes), 3′- and 5′- untranslated regions (UTR) (black boxes) of the *AVPR* gene. The locations of the PCR amplified fragments and the identified mutation are shown. (**G**) Partial DNA sequencing of exon 2 of the *AVPR2* gene in the patient and a control. We identified a novel heterozygous missense mutation (c.493G > C, p.Ala165Pro; noted by an asterisk) in the patient. (**H**) Agarose gel electrophoresis of *Mwo*I-digested PCR fragments. The mutation creates a restriction site for this enzyme, resulting in fragments of 200 base pairs (bp) and 83 bp (not shown), whereas the normal allele results in fragments of 283 bp. The patient is heterozygous for the mutation, the father is hemizygous and the mother is homozygous for the normal allele. We used a 100 bp ladder as a DNA size marker.

**Figure 2 jpm-12-00118-f002:**
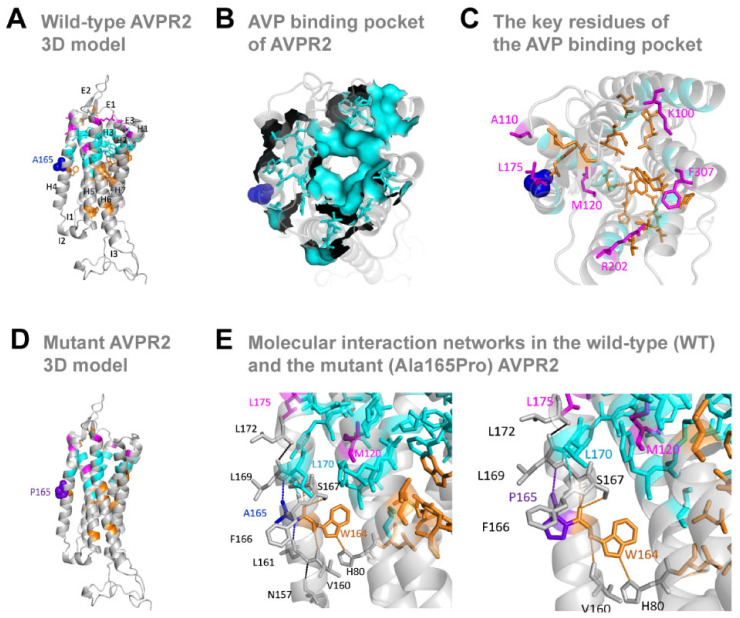
Model structure with key regions of human AVPR2 and Ala165Pro-AVPR2. (**A**) Lateral view of the 3D protein model of the wild-type human AVPR2, including seven transmembrane domains/helices (H1–7) connected with three intracellular (I1–3) and extracellular (E1–3) loops. Dark blue sphere: Alanine residue at position 165 (A165). (**B**) Surface mapping of the vasopressin (AVP) ligand-binding pocket in AVPR2. (**C**) Key residues in the AVP binding region and conserved residues of rhodopsin-like GPCR family. (**D**) Lateral view of the 3D protein model of the mutant Ala165Pro-AVPR2. (**E**) Molecular interaction networks around residue 165 in the WT-AVPR (left panel) and the mutant Ala165Pro-AVPR2 (right panel). Dark blue sphere: Alanine (A) residue at position 165 (A165); Purple blue sphere: Proline (P) residue at position 165 (P165); Cyan sticks and surface: AVP binding pocket; Orange sticks: conserved residues; Magenta sticks: -binding residues; Grey sticks: other residues in the network; Black dash: hydrogen bonds. Residues are indicated by their position number, preceded by the corresponding amino-acid letter: A, Alanine; F, Phenylalanine; H, Histidine; L, Leucine; M, Methionine; N, Asparagine; P, Proline; S, Serine; V, Valine; W, Tryptophan.

**Figure 3 jpm-12-00118-f003:**
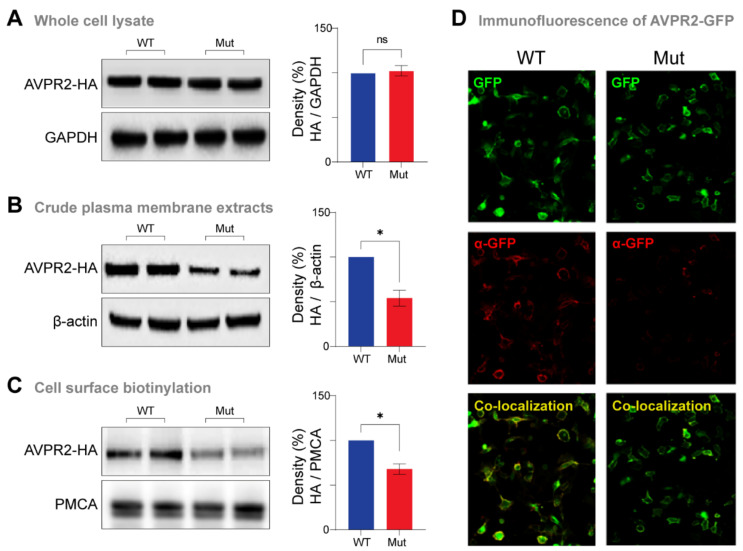
A165P mutant impairs the cell surface localization of AVPR2. (**A**) HEK293 cells were transfected with HA-tagged WT-AVPR2 (WT) or Ala165Pro-AVPR2 (Mut). At 36 h after transfection, cells were harvested, lysed and proteins were analyzed by immunoblotting (left panel). Glyceraldehyde 3-phosphate dehydrogenase (GAPDH) was used as a loading control. Semiquantitative analysis of the effects of the Ala165Pro mutation on the total levels of AVPR2 protein was performed by densitometry (right panel). (**B**) Crude plasma membrane extracts were prepared from cells transfected with HA-tagged WT or Mut AVPR2 and proteins were analyzed by immunoblotting (left panel). β-actin was used as a loading control. A semiquantitative analysis of the effects of Ala165Pro mutation on the plasma membrane levels of AVPR2 protein was performed by densitometry (right panel). (**C**) In parallel, to analyze the cell surface expression of AVPR2, HA-tagged WT or Mut AVPR2 expressing cells were biotinylated using Sulfo-NHS-SS-Biotin and then precipitated with streptavidin-Sepharose beads. Immunoblotting was performed with respective antibodies to visualize the surface expression of proteins (left panel). Plasma membrane Ca^2+^ ATPase (PMCA) was used as a loading control. A semiquantitative analysis of the effects of Ala165Pro mutation on the cell surface abundance levels of AVPR2 protein was performed by densitometry (right panel). (**D**) HEK293 cells were transfected with GFP-tagged WT or Mut AVPR2. At 36 h after transfection, cells were incubated with α-GFP antibody for 1 h at 4 °C followed by cell fixation and incubation with Alexa 547 conjugated secondary antibody. The cellular localization of AVPR2 was visualized through immunofluorescence microscopy (40× magnification). Consistent with the immunoblotting results above, the Mut displays less membrane staining when compared to the WT. Green, GFP staining indicates the intracellular and membrane location; red, α-GFP staining indicates only the membrane location; yellow, co-localization between the green GFP and red α-GFP staining. Values in all bar charts are given in means ± SEM (*n* = 3). Asterisks indicate significant differences (unpaired *t*-tests with Welch’s correction, two-tail): * *p* ≤ 0.05; ns, not significant.

**Figure 4 jpm-12-00118-f004:**
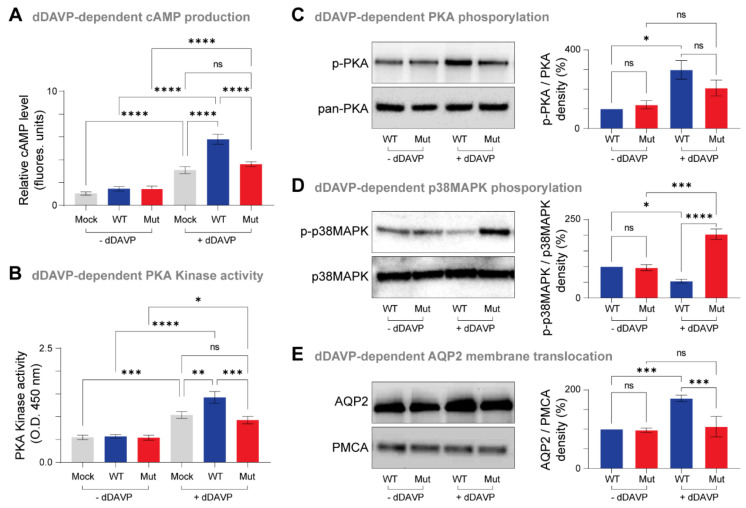
Ala165Pro-AVPR2 mutant impairs the activation of AVP-dependent downstream signaling pathways. (**A**,**B**) HEK293 cells were transiently transfected with WT-AVPR2 (WT) or Ala165Pro-AVPR2 (Mut). After 36 h, the cells were stimulated with dimethyl sulfoxide (DMSO) control (−dDAVP) or 1 μM dDAVP (+dDAVP) for 10 min and processed for quantification of intracellular cAMP **(A)** and PKA kinase activity. (**B**) The Mut AVPR2 displays lower cAMP and PKA activation levels than the WT. (**C**,**D**) WT and Mut AVPR2 expressing cells were stimulated with DMSO control (−dDAVP) or 250 nM dDAVP (+dDAVP) for 60 min. Cells were then harvested, lysed and proteins were analyzed by immunoblotting using antibodies against total and phosphorylated (p-) PKA (**C**) and p38-MAPK (**D**) (left panels). Phosphorylated protein abundance was normalized against the respective total proteins and a semiquantitative analysis of the effects of Ala165Pro phosphorylation was done by densitometry (right panels). (**E**) WT and Mut AVPR2 expressing cells were stimulated with DMSO control (−dDAVP) or 250 nM dDAVP (+dDAVP) for 60 min. Following the treatment, cells were biotinylated using Sulfo-NHS-SS-Biotin and then precipitated with streptavidin-Sepharose beads. Immunoblotting was performed with antibodies against AQP2 to visualize its surface expression (left panel). PMCA was used as a loading control. A semiquantitative analysis of the effects of Ala165Pro mutation on the plasma membrane levels of AVPR2 protein was done by densitometry (right panel). Values in all bar charts are given in means ± SEM (*n* = 3). Asterisks indicate significant differences (one-way ANOVA; Tukey multiple comparisons correction): * *p* ≤ 0.05; ** *p* ≤ 0.01; *** *p* ≤ 0.001; **** *p* ≤ 0.001; ns, not significant.

## Data Availability

The data presented in this study are available in Figure 1, Figure 2, Figure 3 and Figure 4. The raw data presented in this study are available on request from the corresponding author. Requests for materials should be addressed to L.R.S.

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
