# Peer review of "Clinical, Genetic and Functional Characterization of a Novel AVPR2 Missense Mutation in a Woman with X-Linked Recessive Nephrogenic Diabetes Insipidus"

_jpm, 2022, doi:10.3390/jpm12010118_

Round 1

Reviewer 1 Report

No changes needed from my end

Reviewer 2 Report

Comments:

In this study, the author has studied the genetic mutation of AVPR2 gene and its functional consequences in a heterozygous female patient. Author has studied the clinical data of the NDI patient and investigated the effect of mutation on HEK293 cell culture.

Moreover, molecular modelling, transfection studies was also done. Finally, author has investigated the effect of AVPR2 mutant on AVP-dependent downstream signaling pathways.

Major comments:

  • I found the manuscript very confusing with full of data. However, it can neither be considered as a case report nor a scientific article in its present form.
  • One heterozygous female patient data is not sufficient to claim anything.
  • HEK293 cell human embryonic kidney cell culture data could not support the hypothesis because it could not be replaced by human iPSC cells.
  • In-vitro molecular modelling was only based on amino acid replacement of AVPR1, so data is not reliable.

Minor comments:

  • Figure 3 B and figure 3 C –Y-axis heading in densitometry data should be corrected.
  • Method: western blotting – precipitation step for biotinylated protein is missing.

Specific comments:

  • Re-writing of the manuscript.
  • NDI- should be excluded from the topic.

Reviewer 3 Report

General Comment: This is an interesting study of a woman with x-linked congenital NDI. The authors use a variety of methods to investigate the mechanism.

Major Comments: none

Minor Comment: Figure 1 appears after the first paragraph of the methods section.  It would be better if placed in the results section.

Round 2

Reviewer 2 Report

na